# The impact of physical exercise on university students' life satisfaction: The chain mediation effects of general self-efficacy and health literacy

Yu-Peng Ye[1]☯, Guo-You Qin[2]☯, Xinyu Zhang[3], Shan-Shan Han[3], Bo Li[3], Ning Zhou[3], Qi Liu[3], Chen-xi Li[3], Yang-Sheng Zhang[4], Qian-qian Shao[2]*

1 School of physical education, Jing-gang-shan University, Ji'an, China, 2 School of Physical Education, Hanjiang Normal University, Shiyan, China, 3 Institute of Sports Science, Nantong University, Nantong, China, 4 School of Physical Education, Nanjing Xiao zhuang University, Nanjing, China

☯ These authors contributed equally to this work.
* shaoqianqian66@163.com

## Abstract

### Objective

This study aims to explore the impact of physical exercise on university students' life satisfaction and analyses the chain mediation effect of general self-efficacy and health literacy, providing empirical reference and theoretical foundation for the comprehensive enhancement and optimization of students' mental health.

### Method

Based on data from the "China University Student Physical Activity and Health Tracking Survey" (CPAHLS-CS) 2024, the measurement scales used included the Physical Activity Rating Scale (PARS-3), the Satisfaction with Life Scale (SWLS), the General Self-Efficacy Scale (GSES), and the 9-item Short Form Health Literacy Scale (HLS-SF9). A total of 4575 valid samples were analyzed.

### Results

A significant positive correlation was found between physical exercise and life satisfaction (r = 0.137, P < 0.01). The total effect of physical exercise on university students' life satisfaction was significant, with an effect value of 0.045 (95%CI = [0.035, 0.054]). The chain mediation effect of general self-efficacy and health literacy in the relationship between physical exercise and life satisfaction was significant, with an effect value of 0.005 (95%CI = [0.004, 0.006]), accounting for 11.4% of the total effect. The direct effect of physical exercise on life satisfaction had a standardized regression coefficient of 0.001, which was not significant.

**Data availability statement:** All relevant data are within the manuscript and its Supporting Information files.

**Funding:** These funding sources did not provide any input in the design of the study, data collection, data analysis, or the writing of this report. This study was supported by the: 1) Jiangxi Province University Humanities and Social Science Research Youth Project (NO: TY22210) 2) Hanjiang Normal University Key Scientific Research Project (Project Number: XJ2020A03).

**Competing interests:** The authors have declared that no competing interests exist.

**Abbreviations:** Boot LLCI, Lower limits of 95% Bootstrap confidence interval; Boot ULCI, Upper limits of 95% Bootstrap confidence interval; CPAHLS-CS, China University Student Physical Activity and Health Tracking Survey; GSE, General Self-Efficacy Scale; HLS-SF, 9-item Short Form Health Literacy Scale; PARS-3, Physical Activity Rating Scale; SWLS, Satisfaction with Life Scale; SWB, subjective well-being; SD, Standardized deviation; SE, standard error.

## Conclusion

University students' life satisfaction is closely related to physical exercise, general self-efficacy, and health literacy. General self-efficacy and health literacy play a full mediating role in the effect of physical exercise on life satisfaction.

## Introduction

The number of university students in China is vast, and the physical and mental health of students has always been an important issue of concern for both the Chinese government and the public. At a press conference in 2024, the Ministry of Education of China reported that the total enrolment in various forms of higher education in the country was 47.63 million. Life satisfaction (subjective well-being, SWB) is a crucial indicator of an individual's subjective sense of happiness, reflecting their overall evaluation and perception of their own life [1–3]. Previous research has shown that life satisfaction not only affects an individual's psychological health and quality of life [4,5], but is also closely related to personality development [6], academic performance [7], career choices [8], and social adaptability [9]. Chinese university students face challenges such as employment, relationships, and heavy academic burdens, all of which can impact their life satisfaction. Particularly in recent years, the psychological health issues of university students in China have become increasingly prominent [10]. Therefore, conducting dedicated research on the life satisfaction of university students holds great practical significance.

Life satisfaction among university students is influenced by a variety of factors, with physical exercise being a positive lifestyle behavior that significantly contributes to improving both physical and mental health [11] Physical exercise promotes blood circulation, enhances cardiovascular and pulmonary function, improves sleep quality, regulates mood, and reduces symptoms of anxiety and depression, thereby boosting overall health [12–14]. Furthermore, physical exercise can enhance social skills, help students establish good interpersonal relationships, and further promote their psychological well-being [15–17]. Notably, although numerous studies have focused on the positive effects of physical exercise on the physical and mental health of university students, few have explored how physical exercise affects life satisfaction, particularly from the perspective of life quality. Therefore, it is essential to investigate the relationship between physical exercise and life satisfaction among university students in depth, to provide scientific evidence for improving their life satisfaction.

Previous research has explored the relationship between physical exercise and life satisfaction among university students. Early studies mostly employed correlation analysis and found a significant positive correlation between physical exercise and life satisfaction [18,19]. Research by Shang et al. suggested that university students who regularly participate in physical exercise report higher life satisfaction compared to those who are less active [20]. However, as most of these studies adopted cross-sectional designs, it is difficult to establish causal relationships, thus further exploration of the underlying mechanisms between the two is necessary. To better

understand the relationship between physical exercise and life satisfaction, some researchers have introduced the concept of mediating variables [21,22]. A mediating variable is one that serves as a transmission link between the independent and dependent variables. By examining the role of mediators, the specific pathway through which the independent variable affects the dependent variable can be revealed [23]. However, most previous studies have focused on a single mediating variable, failing to comprehensively explore how physical exercise influences life satisfaction through multiple pathways. Therefore, this study will introduce two mediating variables—general self-efficacy and health literacy—and employ a chain mediation effect model to thoroughly examine the mechanisms through which physical exercise affects life satisfaction among university students.

General self-efficacy refers to an individual's confidence in their ability to accomplish specific tasks, and it directly influences their behavioral decisions and performance [**24**]. Previous research supports the idea that general self-efficacy is related to changes in physical activity behavior [25,26]. Wang's study suggests that the impact of physical exercise on university students' life satisfaction is partially mediated by general self-efficacy [27]. Specifically, university students who regularly engage in physical exercise are more likely to develop positive self-cognitions and believe in their ability to overcome challenges, thereby enhancing their life satisfaction. This finding provides new theoretical insights into how physical exercise influences life satisfaction among university students.

Health literacy is a concept that pertains to an individual's ability to obtain, understand, and use basic health information and services to make appropriate health decisions. Health literacy may be an important mediating variable in the relationship between physical exercise behavior and life satisfaction [28–30]. Yang's research found that physical activity and subjective well-being mediate the relationship between physical literacy and health-related quality of life, with a strong association between physical literacy and health literacy [31]. However, there is currently a lack of direct evidence supporting health literacy as a mediating variable.

This study employs a chain mediation effect model to explore the impact of physical exercise on university students' life satisfaction through the chain mediation effect of general self-efficacy and health literacy. Integrating self-determination theory and the health belief model, this study hypothesizes that physical exercise enhances life satisfaction by improving physical health, promoting mental health, enhancing social skills, and increasing health literacy levels. General self-efficacy, as a manifestation of individual autonomy and intrinsic motivation, can increase an individual's willingness and behavior to engage in physical exercise, thereby indirectly improving life satisfaction. Health literacy, as an individual's ability to acquire, understand, evaluate, and apply health information, can foster the development of healthy habits, indirectly boosting life satisfaction. The proposed model is illustrated in Fig 1.

## Research methodology

This study is based on data from the 2024 "Chinese College Students Physical Activity and Health Longitudinal Survey" (CPAHLS-CS). The CPAHLS-CS aims to collect high-quality, individual-level microdata that represents the physical activity and physical and mental health behaviors of Chinese university students. The data is used to analyses the intersection

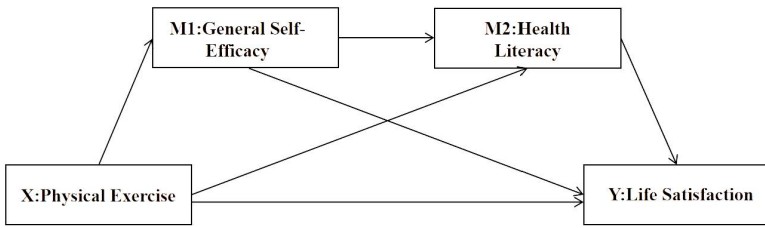

**Fig 1. Hypothetical model diagram.**

of physical activity and health among university students in China, promoting interdisciplinary research on students' health issues. CPAHLS-CS has been widely used in studies analyzing the health status of Chinese university students [32–42]. The participants selected for this study were students from regular higher education institutions in Jiangsu Province, with the list of institutions referring to the Ministry of Education's "National List of Regular Higher Education Institutions (as of June 20, 2024)". **The recruitment period for the study spanned from 08/10/2024 to 09/11/2024, with the subsequent questionnaire survey being conducted from 11/11/2024 to 24/11/2024**. The distribution of research subjects is shown in Table 1.

We employed a questionnaire survey method conducted entirely online. Prior to the completion of the survey, participants were provided with a clear explanation of the study's purpose, intended use of the data, and any incentives for participation. Given that this study involved adult participants and no sensitive or high-risk procedures, the institutional ethics committee at Nantong University determined that written consent was not mandatory. The Study protocol for this study received approval from the ethics committee at Nantong University, documented under approval number 2022(70). Instead, oral informed consent was obtained and documented via the online platform's submission logs, ensuring voluntary participation without compromising ethical standards." Participants were informed that their participation was voluntary and that they could choose not to proceed without any repercussions. This process constituted an oral informed consent, which was documented through the online platform's submission logs. Each participant's agreement to participate was recorded electronically, ensuring a clear audit trail of consent. Given the nature of our research, which did not involve minors, parental or guardian consent was not required. Furthermore, no ethical committee waived the need for consent, as we ensured that all participants were fully informed and voluntarily agreed to participate in the study. This approach adheres to ethical standards by maintaining transparency and respecting participants' autonomy in decision-making.

### Variable measurement

**Physical exercise.** Physical exercise among university students was measured using the Physical Activity Rating Scale (PARS-3), developed by Hiroshi Hashimoto and revised by Liang [43]. PARS-3 assesses an individual's level of physical exercise based on three aspects: intensity, frequency, and duration of each exercise session, to gauge the overall level of exercise participation. The results from the PARS-3 measure the amount of physical exercise, reflecting the students' exercise behaviors during a specific time. Each item in the questionnaire is scored on a 5-point scale, ranging from 1 to 5. The original score calculation formula for PARS-3 is: ***Physical exercise score = intensity × (duration – 1) × frequency***. The test-retest reliability of PARS-3 in this study was 0.820.

**Table 1. Overview of Sample Distribution.**

| Variable | | n | % |
|---|---|---|---|
| Gender | | | |
| | Male | 1187 | 25.9 |
| | Female | 3388 | 74.1 |
| Ethnicity | | | |
| | Han Chinese | 4428 | 96.8 |
| | Ethnic Minorities | 147 | 3.2 |
| Grade | | | |
| | freshman | 2467 | 53.9 |
| | sophomore | 1341 | 29.3 |
| | junior | 643 | 14.1 |
| | senior | 124 | 2.7 |
| | Total | 4575 | 100 |

**Life satisfaction.** Life satisfaction was measured using the Satisfaction with Life Scale (SWLS). Developed by American scholar Diener and colleagues in the 1980s, SWLS consists of five items, with responses scored on a 7-point scale [44]. Higher total scores on the SWLS indicate greater life satisfaction. The applicability of SWLS for Chinese university students has been validated in several studies, demonstrating its suitability for measuring life satisfaction among Chinese students [45,46]. The scale has a reliability of 0.78, CFI of 0.96, GFI of 0.97, and RMSEA of 0.071, the Cronbach's α of 0.89, consistent with prior validation studies. These values suggest it has good reliability and validity, making it a suitable tool for measuring life satisfaction in research.

**General self-efficacy.** General self-efficacy was measured using the General Perceived Self-Efficacy Scale (GSES) [47]. The Chinese version of the GSES was translated and revised by Wang in 2001, and its reliability and validity were analyzed. The GSES showed good reliability, with an internal Cronbach's α = 0.87, a test-retest reliability of r = 0.83 ($P$ < 0.001), and a split-half reliability of r = 0.82 (P < 0.001). The critical threshold for GSES is 2.5; scores below 2.5 indicate low general self-efficacy [48].

**Health literacy.** Health literacy was measured using the 9-item Short Version of the Health Literacy Scale (HLS-SF9) [49,50]. The HLS-SF9 was derived from the HLS-SF12, which was developed by Tuyen [51], specifically, for Asian countries. Based on the HLS-SF12, Sun simplified it into the 9-item version, HLS-SF9. The HLS-SF9 does not exhibit ceiling or floor effects, with a Cronbach's α coefficient of 0.913 and a split-half reliability of 0.871. The concordance level for criterion validity between the HLS-SF9 and HLS-SF12 was ICC = 0.989 (95% CI = 0.988–0.999), indicating that the HLS-SF9 has excellent reliability and validity. It is an effective tool for quickly assessing health literacy in the Chinese population. The specific items are measured on a four-point scale: "very difficult," "difficult," "easy," and "very easy," and each set of three items represents a dimension: healthcare, disease prevention, and health promotion.

## Statistical methods

In this study, data were processed using SPSS 27.0 and Excel software, with the entire process divided into several key stages: 1) Preliminary processing of the data collected via the Wen-juan-xing(V 3.0) platform was conducted using Excel, including re-measuring or removing incomplete or anomalous data. 2) Common method bias detection was performed using SPSS to ensure the accuracy of the research findings. 3) Descriptive analysis was conducted, with chi-square tests used to explore differences in key variables across gender, ethnicity, and academic year. The effect size of these differences was represented by $\eta^2$, where the range is from 0 to 1. Specifically, $\eta^2 < 0.01$ indicates a small effect, $0.01 \leq \eta^2 < 0.06$ indicates a medium effect, and $\eta^2 \geq 0.06$ indicates a large effect. 4) Pearson correlation analysis was used to examine the relationships between the key variables. 5) Regression analysis was employed to test the mediation effects, with the specific analysis of the mediation effects conducted using the Process plugin, selecting Model 6, setting the confidence interval at 95%, and using 5000 bootstrap samples.

## Research results

### Descriptive analysis

Table 2 shows the differences in physical exercise, life satisfaction, general self-efficacy, and health literacy across gender, ethnicity, and academic year. Specifically, about gender, there was a significant difference in physical exercise between male and female students ($\eta^2 = 0.155$). In terms of ethnicity, there were small differences between Han and minority ethnic students in life satisfaction ($\eta^2 = 0.003$), general self-efficacy ($\eta^2 = 0.002$), and health literacy ($\eta^2 = 0.005$). Regarding academic year, there were small differences across different academic years in physical exercise ($\eta^2 = 0.005$) and life satisfaction ($\eta^2 = 0.006$), while there were moderate and significant differences in general self-efficacy ($\eta^2 = 0.010$) and health literacy ($\eta^2 = 0.019$).

**Table 2. Descriptive Analysis Results of Key Variables.**

| | | | Physical Exercise | Life Satisfaction | General Self-Efficacy | Health Literacy |
|---|---|---|---|---|---|---|
| Overall | | | | | | |
| | | M | 15.52 | 23.872 | 1.859 | 28.127 |
| | | sd | 17.911 | 5.617 | 0.539 | 4.189 |
| Gender | | | | | | |
| | Male | | | | | |
| | | M | 27.41 | 24.21 | 1.927 | 28.361 |
| | | sd | 24.747 | 5.896 | 0.591 | 4.726 |
| | Female | | | | | |
| | | M | 11.35 | 23.754 | 1.836 | 28.046 |
| | | sd | 12.32 | 5.512 | 0.518 | 3.981 |
| | F | | 836.279 | 5.79 | 25.4 | 4.97 |
| | P | | <0.001 | 0.016 | <0.001 | 0.026 |
| | η2 | | 0.155 | 0.001 | 0.006 | 0.001 |
| Ethnicity | | | | | | |
| | Han Chinese | | | | | |
| | | M | 15.56 | 23.929 | 1.864 | 28.179 |
| | | sd | 17.936 | 5.612 | 0.54 | 4.195 |
| | Ethnic Minorities | | | | | |
| | | M | 14.33 | 22.17 | 1.725 | 26.578 |
| | | sd | 17.139 | 5.503 | 0.498 | 3.681 |
| | F | | 0.663 | 13.989 | 9.509 | 20.865 |
| | P | | 0.415 | <0.001 | 0.002 | <0.001 |
| | η2 | | <0.001 | 0.003 | 0.002 | 0.005 |
| Grade | | | | | | |
| | freshman | | | | | |
| | | M | 15.11 | 23.563 | 1.828 | 27.893 |
| | | sd | 17.211 | 5.496 | 0.527 | 4.045 |
| | sophomore | | | | | |
| | | M | 14.72 | 24.024 | 1.858 | 27.892 |
| | | sd | 16.85 | 5.498 | 0.524 | 4.178 |
| | junior | | | | | |
| | | M | 18.65 | 24.845 | 1.986 | 29.537 |
| | | sd | 22.144 | 6.243 | 0.59 | 4.456 |
| | senior | | | | | |
| | | M | 15.97 | 23.339 | 1.827 | 28.032 |
| | | sd | 16.617 | 5.239 | 0.563 | 4.298 |
| | F | | 7.954 | 9.66 | 14.826 | 28.781 |
| | P | | <0.001 | <0.001 | <0.001 | <0.001 |
| | η2 | | 0.005 | 0.006 | 0.01 | 0.019 |

## Correlation analysis

The results of the Pearson correlation analysis in Table 3 show significant correlations between all key variables. Physical exercise is significantly positively correlated with general self-efficacy ($r = 0.219$, $P < 0.01$), meaning that university students who engage in more physical exercise tend to have higher general self-efficacy. Physical exercise is also significantly

**Table3. Correlation Analysis Results of Key Variables.**

|  | Physical Exercise | General Self-Efficacy | Health Literacy |
|---|---|---|---|
| General Self-Efficacy | 0.219** |  |  |
| Health Literacy | 0.165** | 0.459** |  |
| Life Satisfaction | 0.137** | 0.589** | 0.400** |

**Table4. Regression Analysis of Variables in the Model.**

| Dependent Variable | Regression Equation Predictors |  | Overall Fit Index | | | Regression Coefficient Significance | | |
|---|---|---|---|---|---|---|---|---|
|  |  |  | R | R2 | F | β | SE | t |
| Life Satisfaction |  |  | 0.607 | 0.368 | 444.208*** |  |  |  |
|  | Physical Exercise |  |  |  |  | 0.001 | 0.004 | 0.049 |
|  |  | General Self-Efficacy |  |  |  | 0.514 | 0.014 | 38.265*** |
|  |  | Health Literacy |  |  |  | 0.162 | 0.018 | 12.137*** |
|  |  | Gender |  |  |  | 0.009 | 0.165 | 0.665 |
|  |  | Ethnicity |  |  |  | −0.021 | 0.376 | −1.783 |
|  |  | Grade |  |  |  | 0.005 | 0.081 | 0.446 |
| General Self-Efficacy |  |  | 0.232 | 0.054 | 64.937*** |  |  |  |
|  | Physical Exercise |  |  |  |  | 0.219 | 0.005 | 13.957*** |
|  |  | Gender |  |  |  | 0.009 | 0.193 | 0.591 |
|  |  | Ethnicity |  |  |  | 0.44 | −0.041 | −2.879** |
|  |  | Grade |  |  |  | 0.062 | 0.095 | 4.312*** |
| Health Literacy |  |  | 0.471 | 0.221 | 259.840*** |  |  |  |
|  | Physical Exercise |  |  |  |  | 0.077 | 0.003 | 5.273*** |
|  |  | General Self-Efficacy |  |  |  | 0.439 | 0.01 | 32.688*** |
|  |  | Gender |  |  |  | 0.028 | 0.136 | 1.958* |
|  |  | Ethnicity |  |  |  | −0.046 | 0.311 | −3.507*** |
|  |  | Grade |  |  |  | 0.054 | 0.067 | 4.106*** |

positively correlated with health literacy (r=0.165, P<0.01), indicating that increased physical exercise is associated with higher levels of health literacy among university students. Additionally, physical exercise is significantly positively correlated with life satisfaction (r=0.137, P<0.01), suggesting that physical exercise has a positive impact on students' life satisfaction. Furthermore, general self-efficacy is significantly positively correlated with health literacy (r=0.459, P<0.01), and general self-efficacy is also significantly positively correlated with life satisfaction (r=0.589, P<0.01). Health literacy is significantly positively correlated with life satisfaction (r=0.400, P<0.01).

## Regression analysis

The results of the regression analysis in Table 4 indicate that in the regression model with life satisfaction as the outcome variable, the overall fit is good ($R^2$=0.368). Both general self-efficacy (β=0.514, P<0.001) and health literacy (β=0.162, P<0.001) have a significant positive effect on life satisfaction. In the regression model with general self-efficacy as the outcome variable, the overall fit is also good ($R^2$=0.054). Physical exercise (β=0.219, P<0.001) has a significant positive effect on general self-efficacy, while ethnicity (β=0.440, P<0.01) and academic year (β=0.062, P<0.001) significantly influence general self-efficacy. In the regression model with health literacy as the outcome variable, the overall fit is relatively good ($R^2$=0.221). Physical exercise (β=0.077, P<0.001) and general self-efficacy (β=0.439, P<0.001) have

a significant positive effect on health literacy, while gender ($\beta=0.028$, $P<0.05$) and academic year ($\beta=0.054$, $P<0.001$) significantly influence health literacy.

In summary, physical exercise significantly and positively predicts general self-efficacy and health literacy, while general self-efficacy and health literacy significantly and positively predict life satisfaction. These findings provide some regression analysis support for the hypothesis that physical exercise influences university students' life satisfaction through the chain mediation effect of general self-efficacy and health literacy.

## Mediation effects

The results of the mediation effect test in Table 5 show that the total effect of physical exercise on university students' life satisfaction is significant, with an effect value of 0.045 (95% CI = [0.035, 0.054]), indicating that physical exercise has an overall positive impact on life satisfaction. The direct effect of physical exercise on life satisfaction is not significant, suggesting that physical exercise may not directly influence life satisfaction.

Regarding the indirect effects: Physical exercise significantly influences life satisfaction through general self-efficacy, with an effect value of 0.035 (95% CI = [0.029, 0.042]), accounting for 79.5% of the total indirect effect. This suggests that general self-efficacy is a more potent mediator than the chain mediation pathway (via both general self-efficacy and health literacy), which accounts for only 11.4%. This finding aligns with Bandura's Social Cognitive Theory, which posits that self-efficacy is a key determinant of behavior, influencing both the initiation and maintenance of physical activities. General self-efficacy, as a broader construct, likely encompasses the confidence individuals have in overcoming barriers to exercise, thus having a stronger impact on life satisfaction. This suggests that general self-efficacy plays an important mediating role between physical exercise and life satisfaction, with physical exercise primarily enhancing life satisfaction by improving general self-efficacy. Furthermore, physical exercise also significantly affects life satisfaction through health literacy, with an effect value of 0.004 (95% CI = [0.002, 0.006]), and the effect size of this path is 9.1%. This indicates that health literacy also mediates the relationship between physical exercise and life satisfaction to some extent. Finally, the chain mediation effect of physical exercise through general self-efficacy and health literacy is significant, with an effect value of 0.005 (95% CI = [0.004, 0.006]), and the effect size accounts for 11.4%. This confirms that physical exercise influences university students' life satisfaction through the chain mediation effect of general self-efficacy and health literacy.

Figure 2 shows that the standardized regression coefficient for the direct effect of physical exercise on life satisfaction is 0.001 and is not significant. This suggests that the impact of physical exercise on university students' life satisfaction is primarily mediated through general self-efficacy and health literacy, rather than through direct effects. In summary, general self-efficacy and health literacy play a chain mediation role in the relationship between physical exercise and life satisfaction, with varying degrees of influence on life satisfaction. Among these, the mediating effect of general self-efficacy is more prominent.

**Table 5. Mediating Effect Tests of Key Variables.**

| Effect | Effect Size | BootSE | 95%CI | | Proportion of Effect |
|---|---|---|---|---|---|
| | | | LLCI | ULCI | |
| Total Effect | 0.045 | 0.005 | 0.035 | 0.054 | |
| Direct Effect | 0.001 | 0.004 | −0.008 | 0.008 | |
| Indirect Effect | 0.044 | 0.003 | 0.038 | 0.051 | |
| Physical Exercise→ General Self-Efficacy→ Life Satisfaction | 0.035 | 0.003 | 0.029 | 0.042 | 79.50% |
| Physical Exercise→ Health Literacy→ Life Satisfaction | 0.004 | 0.001 | 0.002 | 0.006 | 9.10% |
| Physical Exercise→ General Self-Efficacy→ Health Literacy→ Life Satisfaction | 0.005 | 0.001 | 0.004 | 0.006 | 11.40% |

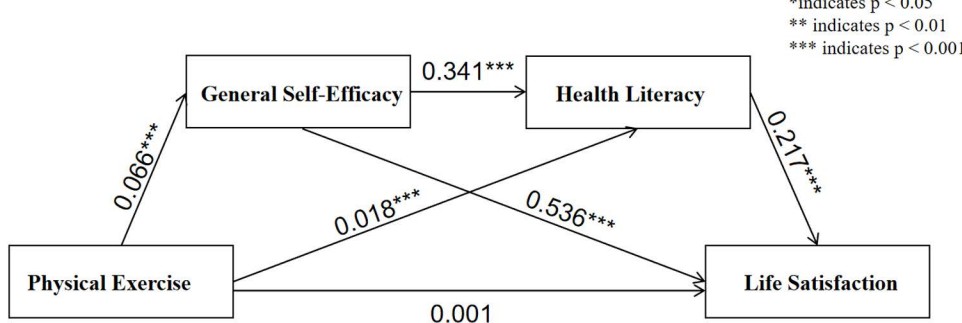

**Fig 2.  The chained mediating effect between physical exercise and life satisfaction.**

## Discussion

This study employed a chain mediation model to explore the mediating roles of general self-efficacy and health literacy in the relationship between physical exercise behavior and life satisfaction among university students. The hypothesis was that physical exercise influences life satisfaction by enhancing an individual's general self-efficacy and health literacy. Specifically, physical exercise enhances individual autonomy and intrinsic motivation, enabling students to approach life's challenges with greater confidence, thereby improving their life satisfaction. Moreover, physical exercise boosts health literacy, helping individuals acquire better knowledge of health, form good habits, and consequently improve their life satisfaction.

The findings of this study reveal a significant positive correlation between physical exercise and life satisfaction among university students, consistent with a wealth of previous research. Simultaneously, the results show that physical exercise has a direct effect on students' subjective well-being [11,27,52]. Physical exercise influences life satisfaction through various physiological and psychological mechanisms, including the release of endorphins, improved sleep quality, and enhanced social support networks [53]. Wiese et al., in their study on the relationship between recreational sports activities and positive emotions, negative emotions, and life satisfaction, found a significant positive correlation between physical exercise and both positive emotions and life satisfaction [54]. The reason physical exercise enhances life satisfaction is that it incorporates factors that contribute to feelings of happiness, such as joy, relaxation, exhilaration, and contentment, while also providing opportunities for self-growth, self-regulation, social recognition, confidence, and self-esteem. The uniqueness of this study lies in the fact that the direct effect of physical exercise on life satisfaction is not significant, and instead, it indirectly influences life satisfaction through the chain mediation effect of general self-efficacy and health literacy. This finding suggests that the impact of physical exercise on psychological well-being may be more dependent on an individual's psychological resources and health management abilities rather than solely on physiological effects.

General self-efficacy plays a significant mediating role in the relationship between physical exercise and life satisfaction [55]. This result aligns with Bandura's self-efficacy theory, which asserts that self-efficacy is an important psychological resource when facing challenges, influencing individuals' behavior choices, effort levels, and persistence. As a positive behavior, physical exercise can enhance general self-efficacy, thus increasing an individual's sense of control over life and their satisfaction with it [56]. McAuley's research found a positive association between self-efficacy and physical exercise [57]. Studies by Hamilton and Chen also support the notion that physical exercise positively predicts an individual's self-efficacy [58,59]. Additionally, research has confirmed that social self-efficacy can predict various life outcomes, including influencing an individual's life goals, social adaptability, and directly affecting life satisfaction [60]. Furthermore, recent studies show that the mediating role of self-efficacy between physical exercise and mental health remains consistent across different cultural contexts [27], further supporting the findings of this study.

Health literacy also plays a significant mediating role between general self-efficacy and life satisfaction. This finding is consistent with Nutbar's health literacy theory, which posits that health literacy not only involves the ability to obtain and understand health information, but also the ability to apply this information to make health-related decisions [61]. Physical exercise enhances general self-efficacy, which in turn boosts health literacy, ultimately improving life satisfaction. This chain mediation effect provides a new perspective on understanding the complex relationship between physical exercise and mental health. Moreover, recent studies suggest that health literacy mediates the relationship between health behaviors and mental health across different populations, further validating the conclusions of this study [62].

This study found that general self-efficacy and health literacy play a mediating role in the effect of physical exercise on life satisfaction. Although the statistical analysis indicates a full mediation model, it is important to note that the magnitude of the total effect (0.045) is relatively small. Therefore, while physical exercise may not have a direct effect on life satisfaction in this model, its influence is primarily exerted through the enhancement of general self-efficacy and health literacy. This result can be explained in two ways: 1) the dominant role of psychological mechanisms. Physical exercise, as an external behavior, exerts its influence primarily through internal psychological mechanisms [63–65]. General self-efficacy, as a core expression of individual autonomy and intrinsic motivation, helps convert the positive experiences of physical exercise into broader confidence and a sense of control over life, thus enhancing life satisfaction. For example, successful experiences in physical exercise (such as achieving a long-distance running goal) can strengthen an individual's belief in their own abilities, which in turn influences their performance in academics, social interactions, and other areas [66–68]. 2) The bridging role of health literacy. Health literacy, as the ability to acquire, understand, and apply health information, converts the health behaviors associated with physical exercise into long-term habits and health management capabilities. By improving health literacy, physical exercise helps university students better cope with health issues, thus indirectly enhancing life satisfaction. For instance, the health awareness developed through physical exercise may prompt individuals to pay more attention to their diet, sleep, and other lifestyle habits, thereby improving overall quality of life [31].

The mediation effects support the applicability of self-determination theory and the health belief model in the relationship between physical exercise and life satisfaction, highlighting the key role of psychological and behavioral cognitive factors. This finding provides new theoretical support for the mechanisms through which physical exercise affects life satisfaction, enriching the research perspective in related fields. Although this study reveals the impact of physical exercise on university students' life satisfaction and its mechanisms, there are still several issues that require further exploration. Firstly, this study examines the overall physical exercise behavior without differentiating the effects of different types, intensities, and frequencies of physical exercise on life satisfaction. Such differentiation is crucial for developing more targeted intervention measures. Additionally, while the statistical results indicate that general self-efficacy and health literacy fully mediate the relationship between physical exercise and life satisfaction, the magnitude of the total effect (0.045) is relatively small. Therefore, caution should be exercised in interpreting this conclusion. Secondly, this study used cross-sectional data, and future research could employ longitudinal designs to explore the causal relationship between physical exercise and mental health, providing a basis for developing long-term psychological health promotion strategies.

## Conclusion and recommendations

The life satisfaction of university students is closely related to physical exercise, general self-efficacy, and health literacy. General self-efficacy and health literacy play a full mediating role in the impact of physical exercise on life satisfaction. The findings of this study provide important practical insights for improving the psychological health of university students. Firstly, universities should recognize the role of physical exercise in promoting psychological health by offering a variety of physical education courses and activities to encourage students to engage in regular exercise. Secondly, universities should focus on enhancing students' self-efficacy and health literacy through mental health education, health knowledge lectures, and other initiatives to strengthen students' psychological resources and health management abilities. Lastly, universities should address the complex relationship between physical exercise and mental health through multi-layered intervention measures, aiming to comprehensively improve students' psychological well-being.

## Supporting information

**S1 Data. S1 Data_Minimum Set.**
(SAV)

## Acknowledgments

We are grateful to the participants and their universities for their cooperation and participation in this study.

## Author contributions

**Conceptualization:** Yu-Peng Ye, Guo-You Qin, Qianqian Shao.

**Data curation:** Yu-Peng Ye, Guo-You Qin, Qianqian Shao.

**Formal analysis:** Yu-Peng Ye, Guo-You Qin, Qianqian Shao.

**Funding acquisition:** Yu-Peng Ye, Qianqian Shao.

**Investigation:** Yu-Peng Ye, Xinyu Zhang, Qianqian Shao.

**Methodology:** Yu-Peng Ye, Xinyu Zhang, Qianqian Shao.

**Project administration:** Yu-Peng Ye, Xinyu Zhang, Qianqian Shao.

**Resources:** Yu-Peng Ye, Shan-Shan Han, Qianqian Shao.

**Software:** Yu-Peng Ye, Shan-Shan Han, Qianqian Shao.

**Supervision:** Yu-Peng Ye, Bo Li, Qianqian Shao.

**Validation:** Yu-Peng Ye, Bo Li, Qianqian Shao.

**Visualization:** Yu-Peng Ye, Ning Zhou, Qianqian Shao.

**Writing – original draft:** Yu-Peng Ye, Ning Zhou, Qi Liu, Chen-xi Li, Yang-Sheng Zhang, Qianqian Shao.

**Writing – review & editing:** Yu-Peng Ye, Qi Liu, Yang-Sheng Zhang, Qianqian Shao.

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
