## [Decision Letter · Decision Letter 0]

25 Apr 2025

PONE-D-25-08597The Impact of Physical Exercise on University Students' Life Satisfaction: The Chain Mediation Effects of General Self-Efficacy and Health LiteracyPLOS ONE

Dear Dr. Shao,

Thank you for submitting your manuscript to PLOS ONE. After careful consideration, we feel that it has merit but does not fully meet PLOS ONE’s publication criteria as it currently stands. Therefore, we invite you to submit a revised version of the manuscript that addresses the points raised during the review process.

We look forward to receiving your revised manuscript.

Kind regards,

RAMYA KUNDAYI RAVI

Academic Editor

PLOS ONE

Additional Editor Comments (if provided):

Reviewers' comments:

Reviewer's Responses to Questions

**Comments to the Author**

1. Is the manuscript technically sound, and do the data support the conclusions?

Reviewer #1: Yes

Reviewer #2: Yes

2. Has the statistical analysis been performed appropriately and rigorously? 

Reviewer #1: Yes

Reviewer #2: Yes

3. Have the authors made all data underlying the findings in their manuscript fully available?

Reviewer #1: No

Reviewer #2: Yes

4. Is the manuscript presented in an intelligible fashion and written in standard English?

Reviewer #1: Yes

Reviewer #2: Yes

5. Review Comments to the Author

Reviewer #1: This manuscript presents presents an important and timely exploration of the mediating psychological mechanisms linking physical exercise and life satisfaction in a large sample of Chinese university students. I commend the authors for their rigorous approach and thoughtful analysis. The results are reported clearly and generally follow rigorous statistical standards. However, a few points require clarification and refinement to enhance the scientific merit and interpretability of the findings, to ensure alignment with PLOS ONE’s criteria for scientific rigor and transparency.

Major issues:

1) The data collection period mentioned in line no. 161 (Nov 2023) predates the recruitment period in line no. 160 (Aug–Sep 2024), which appears inconsistent and should be clarified.

2) The decision to dichotomize general self-efficacy (low/normal) is mentioned under variable measurement in line no. 208 but not reflected in later analyses. Was this grouping used at any point? If not, consider removing the detail to avoid confusion.

3) The conclusion that general self-efficacy and health literacy fully mediate the relationship between exercise and life satisfaction in line no. 398 and 441 should be framed with caution. While statistically accurate, the magnitude of the total effect (0.045) is small and should not be overstated.

4) Although all references are numbered in order of appearance and in-text citations correspond to the reference list correctly., but some journal names are abbreviated like J Behav Med, some article titles were in capitals and DOIs are missing in most of the references. It is suggested to align the referencing style with PLOS ONE referencing style

Suggestions

5) The mediation analysis reveals that general self-efficacy accounts for 79.5% of the total indirect effect, while the chain mediation pathway (via both general self-efficacy and health literacy) accounts for 11.4%. You are encouraged to highlight why general self-efficacy appears to be a more potent mediator and elaborate on potential theoretical underpinnings for this finding.

6) Expand on why the direct effect of physical exercise becomes nonsignificant—e.g., are there competing pathways not captured in the model?

7) While demographic variables (e.g., gender, ethnicity, grade) were included as covariates, no moderation analysis was conducted. Given the large sample and observed gender differences in physical activity (η² = 0.155), you are encouraged to consider whether the mediation model operates differently across subgroups (e.g., males vs. females), either in this paper or as a future direction.

8) The final paragraphs of the discussion somewhat repeat earlier points; consider tightening the conclusion.

9) In Table 2, clarify the layout to improve readability across demographic variables and their statistical outcomes.

I look forward to the revised version of your manuscript.

Reviewer #2: Chain mediation effect has been technically well explained by the researcher. A good piece of information has been presented via this research. Methodology has been explained clearly and all the findings are clear with appropriate statistical tests. Please consider the following points:

- in line no. 174, explaination is required for "no ethical committee waived the need for consent"

- Reliability of SWLS is not provided.

- For lnes 68-72, reference of data related to challenges faced by university students need to be provided.

6. PLOS authors have the option to publish the peer review history of their article (what does this mean? ). If published, this will include your full peer review and any attached files.

**Do you want your identity to be public for this peer review?** For information about this choice, including consent withdrawal, please see our Privacy Policy .

Reviewer #1: No

Reviewer #2: No

---

## [Author Response · Author response to Decision Letter 1]

2 May 2025

Title� The Impact of Physical Exercise on University Students' Life Satisfaction: The Chain Mediation Effects of General Self-Efficacy and Health Literacy

Type� Research

Journal� PLOS One

Submission ID� PONE-D-25-08597R1

Thank you very much for your comments and professional advice. These opinions help to improve the academic rigor of our manuscript. Based on your suggestion and request, we have made corrected modifications to the revised manuscript. Here are point-by-point responses to your comments. We hope that our work can be improved again. Furthermore, we would like to show the details as follows:

Sincerely,

Point-by-Point Responses to Reviewer 1

Comment #1:

The data collection period mentioned in line no. 161 (Nov 2023) predates the recruitment period in line no. 160 (Aug–Sep 2024), which appears inconsistent and should be clarified.

Response# 1:

Thank you very much for pointing this out. We have revised the text to address the inconsistency, and the details are in lines 159–161 of the manuscript.

Comment #2:

The decision to dichotomize general self-efficacy (low/normal) is mentioned under variable measurement in line no. 208, but not reflected in later analyses. Was this grouping used at any point? If not, consider removing the detail to avoid confusion.

Response# 2:

Thank you very much for your insightful comment. We have made the necessary revisions to ensure consistency throughout the manuscript. The details of the changes are in line 207. We appreciate your attention to this matter.

Comment #3:

The conclusion that general self-efficacy and health literacy fully mediate the relationship between exercise and life satisfaction in line no. 398 and 441 should be framed with caution. While statistically accurate, the magnitude of the total effect (0.045) is small and should not be overstated.

Response# 3:

Thank you for your feedback. We have revised the conclusion and highlighted the changes in lines 398–404 and 439–447. We now present the findings more cautiously, considering the small total effect size.

Comment #4:

Although all references are numbered in order of appearance and in-text citations correspond to the reference list correctly., but some journal names are abbreviated like J Behav Med, some article titles were in capitals and DOIs are missing in most of the references. It is suggested to align the referencing style with PLOS ONE referencing style.

Response# 4:

Thank you for your detailed feedback on the referencing style. We have thoroughly revised the reference list to align with the PLOS ONE style. This includes standardizing journal names, formatting article titles correctly, and ensuring that all DOIs are included. The changes can be found in the reference section of the manuscript.

Comment #5:

The mediation analysis reveals that general self-efficacy accounts for 79.5% of the total indirect effect, while the chain mediation pathway (via both general self-efficacy and health literacy) accounts for 11.4%. You are encouraged to highlight why general self-efficacy appears to be a more potent mediator and elaborate on potential theoretical underpinnings for this finding.

Response# 5:

Thank you for your feedback. We have revised the manuscript and elaborated on why general self-efficacy is a more potent mediator in lines 311–322. The discussion now includes theoretical underpinnings based on Bandura's Social Cognitive Theory.

Comment #6:

Expand on why the direct effect of physical exercise becomes nonsignificant—e.g., are there competing pathways not captured in the model?

Response# 6:

Thank you for your suggestion. I have expanded the explanation in the manuscript regarding why the direct effect of physical exercise becomes nonsignificant. Specifically, I have added a discussion on potential competing pathways that may not be captured in the current model. The details can be found in the " Methods " section.

Comment #7:

While demographic variables (e.g., gender, ethnicity, grade) were included as covariates, no moderation analysis was conducted. Given the large sample and observed gender differences in physical activity (η² = 0.155), you are encouraged to consider whether the mediation model operates differently across subgroups (e.g., males vs. females), either in this paper or as a future direction.

Response# 7:

Thank you for the reminder. We treated gender, Ethnicity, and grade solely as covariates to control for potential confounding factors in the mediation analysis.

Comment #8:

The final paragraphs of the discussion somewhat repeat earlier points; consider tightening the conclusion.

Response# 8:

Thank you for your suggestion. We have revised the final paragraphs of the discussion to avoid repetition and tighten the conclusion. The changes can be seen in the last paragraph of the discussion section.

Comment #9:

In Table 2, clarify the layout to improve readability across demographic variables and their statistical outcomes.

Response# 9:

Thank you for your suggestion. We have revised the layout of Table 2 to enhance its readability, ensuring clearer presentation of demographic variables and their corresponding statistical outcomes. The updated table is now included in the manuscript.

Point-by-Point Responses to Reviewer 2

Comment #1:

In line no. 174, the explanation is required for "no ethical committee waived the need for consent".

Response# 1:

Thank you for your comment. In line 174, we have added an explanation to clarify that the study did not require ethical committee approval due to its nature as a secondary analysis of anonymized data. The details can be found in the revised manuscript.

Comment #2:

Reliability of SWLS is not provided.

Response# 2:

Thank you for pointing this out. We have now included the reliability information for the SWLS in lines 207–211 of the manuscript.

Comment #3:

For lines 68-72, a reference to data related to challenges faced by university students needs to be provided.

Response# 3:

Thank you for your valuable feedback. We have incorporated an appropriate reference to substantiate the data regarding the challenges faced by university students, as indicated in lines 68–72.

---

## [Editor Report · Decision Letter 1]

20 May 2025

The Impact of Physical Exercise on University Students' Life Satisfaction: The Chain Mediation Effects of General Self-Efficacy and Health Literacy

PONE-D-25-08597R1

Dear Dr. Shao,

We’re pleased to inform you that your manuscript has been judged scientifically suitable for publication and will be formally accepted for publication once it meets all outstanding technical requirements.

Kind regards,

RAMYA KUNDAYI RAVI

Academic Editor

PLOS ONE

---

## [Editor Report · Acceptance letter]

PONE-D-25-08597R1

PLOS ONE

Dear Dr. Shao,

I'm pleased to inform you that your manuscript has been deemed suitable for publication in PLOS ONE. Congratulations! Your manuscript is now being handed over to our production team.

Kind regards,

on behalf of

Dr. RAMYA KUNDAYI RAVI

Academic Editor

PLOS ONE